# Nutriepigenomics in Environmental-Associated Oxidative Stress

**DOI:** 10.3390/antiox12030771

**Published:** 2023-03-21

**Authors:** Karla Rubio, Estefani Y. Hernández-Cruz, Diana G. Rogel-Ayala, Pouya Sarvari, Ciro Isidoro, Guillermo Barreto, José Pedraza-Chaverri

**Affiliations:** 1International Laboratory EPIGEN, Consejo de Ciencia y Tecnología del Estado de Puebla (CONCYTEP), Instituto de Ciencias, Ecocampus, Benemérita Universidad Autónoma de Puebla (BUAP), Puebla 72570, Mexico; 2Laboratoire IMoPA, Université de Lorraine, CNRS, UMR 7365, F-54000 Nancy, France; 3Lung Cancer Epigenetics, Max-Planck-Institute for Heart and Lung Research, 61231 Bad Nauheim, Germany; 4Postgraduate in Biological Sciences, Universidad Nacional Autónoma de México, Ciudad Universitaria, Ciudad de Mexico 04510, Mexico; 5Departamento de Biología, Facultad de Química, Universidad Nacional Autónoma de México, Av. Universidad 3000, Ciudad de Mexico 04510, Mexico; 6Independent Researcher, Puebla 72570, Mexico; 7Department of Health Sciences, Università del Piemonte Orientale, Via Paolo Solaroli 17, 28100 Novara, Italy

**Keywords:** DNA methylation, ncRNAs, histone modifications, nutrition, antioxidants, 2D culture, extracellular vesicles

## Abstract

Complex molecular mechanisms define our responses to environmental stimuli. Beyond the DNA sequence itself, epigenetic machinery orchestrates changes in gene expression induced by diet, physical activity, stress and pollution, among others. Importantly, nutrition has a strong impact on epigenetic players and, consequently, sustains a promising role in the regulation of cellular responses such as oxidative stress. As oxidative stress is a natural physiological process where the presence of reactive oxygen-derived species and nitrogen-derived species overcomes the uptake strategy of antioxidant defenses, it plays an essential role in epigenetic changes induced by environmental pollutants and culminates in signaling the disruption of redox control. In this review, we present an update on epigenetic mechanisms induced by environmental factors that lead to oxidative stress and potentially to pathogenesis and disease progression in humans. In addition, we introduce the microenvironment factors (physical contacts, nutrients, extracellular vesicle-mediated communication) that influence the epigenetic regulation of cellular responses. Understanding the mechanisms by which nutrients influence the epigenome, and thus global transcription, is crucial for future early diagnostic and therapeutic efforts in the field of environmental medicine.

## 1. Introduction

The epigenome encompasses multiple interacting regulatory elements that define phenotypic variation beyond what the DNA sequence encodes [1,2,3]. In addition to their roles in development and differentiation, epigenetic mechanisms play a fundamental role in transcriptional regulation during disease progression, further underscoring the importance of understanding their molecular bases [4,5]. Chromatin is a complex of proteins and nucleic acids found in the eukaryotic cell nucleus and is subjected to continuous changes to accommodate cellular metabolic needs. As chromatin is the physiological template of the majority of the epigenetic mechanisms, it refers to heritable and reversible changes that mediate all DNA-dependent processes, including replication, repair, recombination and transcription, without altering the DNA sequence. In a versatile and highly coordinated manner, epigenetic regulation of transcription establishes cell-specific gene expression signatures from the same genome. It allows the cell to change these gene expression signatures in response to stimuli, such as changing conditions from their environment [2]. Some of the best-characterized epigenetic mechanisms of transcriptional regulation involve DNA methylation, histone modifications, nucleosome remodeling, regulation via noncoding RNAs (ncRNAs) and nuclear matrix interactions [5,6,7,8].

## 2. Epigenetics in the Context of Environmental Exposure

DNA methylation in eukaryotes is mediated by DNA methyltransferases (DNMTs) and refers to the covalent transfer of a methyl group to the C5 position of cytosine forming 5-methylcytosine (5mC), most frequently at the dinucleotide sequence CG (mCG) [9]. DNA regions that are ≥200 bp long and show a CG:GC ratio ≥0.6 are defined as CpG (citosine-phosphate-guanosine) islands (CGIs), which are often located within the promoter of protein-coding genes [10]. It is noted that methylation reprogramming can result from the inhibition of DNMTs or de novo DNMTs activity. This mechanism is reversible and can be mediated by different mechanisms involving DNA repair and ten-eleven translocation (TET) enzymes, which catalyze the oxidation of 5mC, forming several intermediates such as 5-hydroxymethylcytosine (5hmC), 5-formylcytosine (5fC) and 5-carboxylcytosine (5caC) until its full conversion to cytosine [11,12]. Although 5hmC, 5fC and 5caC are less abundant than 5mC, they contribute to a sensitive and dynamic read-out of cell state as their profiles are in part determined by an active gene–body transcription and enhancer activity, which are rapidly altered upon environmental challenge [13].

The genomic landscape of 5mC is the product of the constant activity of both DNA methylation and demethylation processes, resulting in a dynamic equilibrium that can be shifted in response to stimuli, including changes in the external environment of the cell. While the loss of 5mC is associated with genomic instability and cancer, the gain of 5mC has been associated with several congenital defects and other diseases, thus highlighting not only the importance of the balance of DNA methylation but also the possible consequences of their alteration [14,15,16]. Changes in nutritional status and environmental exposure to several agents can modify genomic DNA methylation patterns, thereby affecting chromatin structure and gene expression toward disease [15,17]. Patterns of 5hmC directly correlate with gene expression profiles and can be rapidly altered following short-term exposure to environmental agents, in a proportional manner to the duration of the exposure. For instance, several 5hmC changes induced by the nongenotoxic carcinogen phenobarbital (PB) were shown to persist in PB-driven tumors; thus, such changes represent early exposure biomarkers that are maintained across cellular progenies [13]. Interestingly, loci enriched with 5hmC after exposure to PB correspond mostly to enhancers and in less quantity to promoters and gene bodies. Furthermore, PB-induced aberrant levels of 5hmC have been associated with liver tumors in mice, therefore suggesting that PB exerts its pathological effect through epigenetic regulation, leading to disease [18,19].

Posttranslational modifications of histone proteins (further referred to as histone modifications) constitute another level of epigenetic mechanisms of transcriptional regulation. Histone proteins (H1, H2A, H2B, H3 and H4) are relatively small and basic proteins that are abundant in the cell nucleus and are an essential part of the nucleosome. The nucleosome is the basic repeating structural and functional unit of chromatin, consisting of the nucleosome core particle, the linker DNA between two nucleosome core particles and an H1 molecule [20]. The nucleosome core particle consists of approximately 146 base pairs of genomic DNA wrapped around a histone octamer of two H2A-H2B dimers and one (H3-H4)2 tetramer [21]. Due to the structural features of the nucleosome, histone proteins can undergo posttranslational modifications at their N-terminal tails, which comprise acetylation, methylation, phosphorylation, ubiquitination and sumoylation, among others [22,23,24,25]. While DNA methylation is relatively stable in somatic cells, histone modifications are more diverse and dynamic, changing rapidly during the cell cycle [6,8,22,23]. Acetylation at specific amino acids of histones (e.g., histone 3 lysine 9 acetylation, H3K9Ac) is generally associated with active chromatin. It is mediated by histone acetyltransferases (HATs) and removed by histone deacetylases (HDACs). Histone methylation also occurs at specific amino acids of histone proteins and can be associated with both repression (e.g., H3 lysine 27 trimethylation, H3K27me3) or activation (e.g., H3 lysine 4 trimethylation, H3K4me3) of gene expression. Various enzymes mediate histone methylation (histone methyltransferases; HMTs) and histone demethylation, represented mainly by histone lysine demethylases (KDMs) [26,27].

Several environmental agents induce changes in histone modifications, thereby leading to changes in gene expression signatures. It has been reported that tobacco smoke influences histone methylation patterns. Further, the cadmium contained in tobacco smoke has been reported to have an impact on epigenetic landscapes in fetuses from smoking mothers. Similarly, exposure to lead (Pb) has been reported to have an impact on neurological disease via alterations in epigenetic regulation [28,29,30]. On the other side, epigenetic modifications require metabolites as substrates. S-adenosylmethionine (SAM) is a common donor of methyl groups. The sources of SAM can be obtained from several metabolic routes such as glycolysis, amino acid metabolism, as well as folate and choline. It has been observed that a reduced methionine intake is associated with a decrease in methylation levels. In addition, the demethylation process requires α-ketoglutarate as a substrate. Furthermore, acetylation processes require acetyl-CoA as a substrate. When there is a high availability of acetyl-CoA, an increase in histone acetylation levels is also observed. The evidence strongly suggests that epigenetic regulation is closely related to nutritional factors [31,32,33]. Perhaps one of the best-known examples of the effects of nutrition on epigenetics and health is the Dutch famine. In a harsh winter during World War II, caloric intake was severely reduced from what is recommended as healthy for human beings. Fetuses gestated under these circumstances showed as adults a higher incidence of obesity, type 2 diabetes, cardiovascular diseases, a propensity to dyslipidemias and even mental disorders. Remarkably, individuals exposed to these conditions showed differentially methylated loci in their genome, known as malnutrition-associated differentially methylated regions (P-DMRs). Such P-DMRs are frequently located in regulatory elements and particularly in regions associated with birth weight and LDL-cholesterol levels [29,34].

Another component of epigenetic mechanisms that plays a significant role in mediating transcription involves ncRNAs. The majority of RNAs transcribed from the mammalian genomes are ncRNAs, which are not translated into proteins [35]. ncRNAs can be classified based on their nucleotide length into small noncoding RNAs (sncRNAs, 21–34 nucleotides long) and long noncoding RNAs (lncRNAs, >200 nucleotides long) [36,37]. Further classification of the ncRNAs can be done based on the biological processes in which they are involved, leading to a variety of ncRNA subtypes, including short interfering RNAs (siRNAs), transfer RNAs (tRNAs), Y RNAs, PIWI-interacting RNAs (piRNAs) and microRNAs (miRNAs) in the fractions spanning 15–40 nucleotides, while ribosomal RNAs (rRNAs) and lncRNAs are predominant in the fractions spanning ≥40 nucleotides [38]. Epigenetic regulation of gene expression signatures by ncRNAs has been mainly related to lncRNAs and miRNAs. miRNAs constitute an RNA subtype of 21 to 25 nucleotides in length that act primarily in the cytosol by inhibiting translation [39]. However, accumulating reports in the last decade demonstrate the presence of functional miRNAs in the cell nucleus as regulators of various biological processes, including transcription [7,40,41,42,43,44]. For lncRNAs, different mechanisms have been characterized in detail for their roles in the nucleus [35,45,46,47,48]. Through their interaction with specific genomic regions and proteins, they tether their regulated genes to specialized regions of the nucleus in which specific biological processes take place. For example, ncRNAs provide a framework for the assembly of defined chromatin structures at specific loci, thereby modulating gene expression, centromere function and the silencing of repetitive DNA elements. Exposure to diverse environmental factors also influences the epigenetic mechanism of the regulation of gene expression mediated by ncRNAs. Recent evidence suggests that heavy metals carry out their toxic action through miRNAs, particularly related to altered epigenetic mechanisms of gene expression in neurological diseases. Lead (Pb) and cadmium (Cd) are good examples of heavy metals that have been associated with the development of Alzheimer’s disease (AD), Parkinson’s disease (PD) and amyotrophic lateral sclerosis (ALS) [28,30,49]. From a nutritional perspective, it has been described that glucose intake activates thioredoxin-interacting protein (TXNIP), which induces *miR-204* expression. In turn, *miR-204* targets MAFA, a key transcription factor for insulin production, contributing to type 2 diabetes mellitus development [50].

## 3. Relationship between Epigenetics and Oxidative Stress in the Context of Environmental Exposure

Exposure to different environmental factors, including pollutants, can change the epigenome and lead to adverse health effects. Pollutants such as heavy metals, endocrine disruptors (EDC), particulate matter (PM) and titanium oxide (TNM), among others, have been linked to epigenetic changes, including DNA methylation, histone modifications and ncRNA aberrant expression [51,52]. However, oxidative stress (OS) is arguably the most common mechanism in the toxicology of environmental agents, unifying the action of broad classes of disparate environmental physicochemical pollutants, including oxidant gases, organic compounds, particulate surfaces and metal ions [53].

OS results from environmental disturbance, which is known to modify cellular processes such as apoptosis, signal transduction cascades and DNA repair mechanisms [54]. As OS is a natural physiological process where the presence of oxidants (reactive oxygen-derived species, ROS and nitrogen-derived species, RNS) overcome the uptake strategy of antioxidant defenses, it plays an essential role in epigenetic changes induced by environmental pollutants and culminates in signaling the disruption of redox control (Figure 1) [55,56].

It is noted that mitochondria are the primary intracellular source of ROS generation due to electron transfer during adenosine triphosphate (ATP) production [57,58,59]. Complexes I and III of the electron transport chain (ETC) have been described as important sites of ROS production [60,61,62]. These sites within ETC can leak even under normal conditions. The electron leak from the ETC reduces molecular oxygen (O_2_) to superoxide anion (O_2_^−^), which triggers the production of hydrogen peroxide (H_2_O_2_), which, in turn, can receive another electron and form the hydroxyl radical (-OH) [63]. On the other hand, O_2_^−^ can also react with nitric oxide (NO), generating peroxynitrite anion (ONOO^−^) [64]. ONOO^−^ and OH are highly reactive ROS that induce oxidative damage to proteins, lipids and DNA when their production is exceeded [65]. Therefore, to reduce the cellular damage caused by ROS, a cellular redox balance is needed. This balance is achieved by different antioxidants, including enzymes such as superoxide dismutase (SOD), glutathione peroxidase (GPx), catalase (CAT) and nonenzymatic antioxidants such as glutathione (GSH) and vitamins A, C and E [66]. The expression of antioxidant enzymes and those associated with GSH production depends on different transcription factors, such as nuclear-erythroid-factor-related factor 2 (NRF2) and hairpin box O (FOXO). They both respond rapidly to oxidative environments to induce a redox balance, such as the expression of the abovementioned antioxidant enzymes [66].

### 3.1. DNA Methylation and Oxidative Stress

Various studies have shown that environmental pollutants alter DNA methylation by inducing OS. Environmental factors, especially plastic-derived EDCs (bisphenol A, BPA; phthalate di-2-Ethylhexyl phthalate, DEHP; and bisphenol-A bis-diphenyl phosphate, BDP), generate OS and induce transgenerational epigenetic modifications through abnormal DNA methylation involving male and female gametes [67,68,69]. OS was previously shown to be one of the mechanisms by which BPA causes alteration in DNA methylation, affecting development and fertility function in male rat pups and grass carp ovary (GCO) cells [70,71]. In addition, prenatal and ancestral exposure to DEHP has been shown to disrupt DNA methylation in the ovaries of CD-1 mice in each generation by altering the activity of enzymes such as DNA methyltransferases (DNMT) and ten-eleven translocation (TET), thus changing gene expression in several pathways critical to ovarian cell growth, proliferation and function. These pathways include the sex steroid hormone synthesis pathway, the phosphoinositide 3-kinase (PI3K) pathway, cell cycle regulators, apoptosis and OS factors, steroid hormone receptors and insulin-like growth factors [72]. Moreover, embryonic exposure to DBP causes hypomethylation of genes involved in heart development, which induces congenital cardiac defects [73].

Cigarette smoke is another pollutant that induces high levels of ROS, which leads to alterations in DNA methylation and the development of diseases such as chronic obstructive pulmonary disease (COPD) in humans [74]. Squamous metaplasia of the respiratory epithelia occurring in tobacco smokers is a typical outcome of such epigenetic changes [75]. Furthermore, growing evidence reveals that environmental exposure to heavy metals involves alterations in DNA methylation [76]. For instance, Cd pollution in the Lean River was observed to cause OS and a significant increase in global DNA methylation in the zebrafish liver [77]. Studies have consistently shown that Cd load in fish, in consequence, is an important health hazard due to human consumption due to an increase in the levels of this toxic element (and others such as Pb and MeHg) from 1 × 10^−6^ to 1 × 10^−3^, correlated with an exponential increase in cancer risk from 1 case per million to 1 case in a thousand individuals, which clearly highlights the necessity of environmental regulations worldwide [78]. Cd also suppresses the activity of the tumor suppressor p16 through hypermethylation of its gene in rats exposed to cadmium chloride (CdCl_2_) [79]. In addition, exposure to environmental arsenic is prominently related to hypomethylation in blood DNA [80]. Using a similar approach, another study showed that environmental Pb significantly decreased *LINE-1* promoter gene methylation, which resulted in pathological consequences in workers from a battery plant [81]. Furthermore, tetranitromethane (TNM) exposure has been shown to cause reduced global DNA methylation while increasing ROS and malondialdehyde (MDA) [82]. Therefore, TNM treatment well explains the association of oxidative blockade with hypermethylation of poly(ADP-ribose) polymerase-1 (*PARP-1*), a sealed DNA-binding protein that regulates various cellular mechanisms [83].

PM2.5 is another environmental factor capable of inducing epigenetic modifications via pro-oxidant activity [84,85,86]. PM2.5 has been shown to increase ROS production through upregulation of the aryl hydrocarbon receptor (*AHR*) in skin keratinocytes. Such ROS increase was related to hypomethylation of the *p16INK4A* promoter due to the downregulation of DNMTs and the upregulation of TET enzymes. Of note, the antioxidant N-acetylcysteine (NAC) could reverse the epigenetic modifications induced by PM2.5, which exposes the role of OS in mediating such an effect [84]. Another study found that PM exposure increased ROS levels and concomitantly activated stress kinase cascades to inhibit p16 transcription in alveolar epithelial cells, both in vivo and in vitro. The decreased expression of *p16* resulted in the loss of cyclin D-dependent kinases (CDK4 and CDK6) inhibition and cell cycle arrest in the G1 phase. Thus, PM-induced DNA methylation of the tumor suppressor *p16INK* could favor the development and progression of lung cancer [56]. In addition, PM-induced ROS produces 5-hmC and causes DNA demethylation, suggesting a significant correlation between oxidative stress, methylation processes and epigenetic alterations (Figure 1) [87].

There are different mechanisms by which OS causes alterations in the methylome [88,89]. One of them is the formation of DNA lesions. For example, -OH generation causes DNA damage such as base modifications, deletions, strand breaks and chromosomal rearrangements [90,91]. These DNA lesions caused by -OH interfere with DNMT activity since the damaged DNA cannot be used as a substrate for the enzyme, resulting in global hypomethylation [92]. On the other hand, ROS can also cause hypomethylation by oxidizing DNA bases in CpG, cytosine-guanosine dinucleotide (CG) DNA sequences where cytosine is the preferred base for DNA methylation and guanine is the site of oxidative damage. Some of the ROS, such as -OH, can oxidize guanine forming mainly (but not exclusively) 8-oxoguanosine (8-oxoG) [93,94,95]. The formation of 8-oxoG decreases the ability of DNMT to methylate an adjacent cytosine [96,97]. The formation of 8-oxoG also causes the N7 position of guanine to become a hydrogen bond donor rather than a hydrogen bond acceptor, substantially decreasing the binding of methyl-CpG-binding proteins (MBPs) [98,99]. Furthermore, if the cytosine next to the oxidized guanine is methylated in the CpG sequence, forming 5-mC, it becomes more susceptible to oxidation by TET, generating 5-hmC [100]. It is noteworthy to mention that oxidation of 5-mC to 5-hmC also causes a reversal of the binding affinity for MBPs [99]. MBPs interact with methylated DNA to drive gene expression and maintain or alter DNA architecture [101]. Therefore, by impairing the MBP–DNA interactions, oxidation of 5-mC may result in hereditary epigenetic alterations.

OS also directly blocks DNA methylation by depleting S-adenosylmethionine (SAM) and oxidizing methionine adenosyltransferase (MAT) and cobalamin in methionine synthase (MS), leading to the inactivation of SAM and MS enzymes [102,103,104]. SAM is formed by MAT from methionine and is the donor of the methyl group used by DNMTs to methylate cytosine. After donating the methyl group to cytosine, SAM becomes S-adenosylhomocysteine (SAH), which hydrolyzes to homocysteine. Homocysteine can be regenerated to methionine by the action of MS. Furthermore, the trans-sulfuration pathway to regenerate glutathione under OS conditions causes the transformation of SAM to homocysteine, cystathionine and cysteine, thus eventually depleting the reservoir of SAM [105,106]. Lastly, OS can also induce DNA hypermethylation by inhibiting TET proteins. TETs are iron- [Fe](II) and α-ketoglutarate- (α-KG)-dependent dioxygenases responsible for 5-mC oxidation. During the TET catalytic cycle, Fe(II) is oxidized to Fe(III) and Fe(IV) [107]. The oxidized iron is regenerated by ascorbate [108]. However, under OS conditions, reduced ascorbate decreases and, consequently, TET is inactivated, increasing overall methylation levels [109].

### 3.2. Histone Modifications and Oxidative Stress

Histones that are part of chromatin nucleosomes can be altered by OS, which is induced by environmental factors [110,111]. Histones can undergo posttranslational modifications such as acetylation, methylation, phosphorylation, nitration, ribosylation, ubiquitination, sumoylation or glycosylation [112,113]. Various studies have shown that OS and nitrosative stress extensively modify histones, affecting their folding and stability and their posttranslational modifications [109]. For example, RNS like ONOO^−^ can use nitrated histones. Nitrated histones show an increase in structured domains, specifically β-sheet structures, and greater thermostability. The increase in thermostability has been related to protecting DNA from OS. Furthermore, in vivo nitration-denitration of histones could be involved in the control of cellular events, including apoptosis [114,115]. The effects of nitration by environmental contaminants have been depicted in cancer models. For instance, in mice with fibrosarcoma caused by methylcholanthrene, a polycyclic aromatic hydrocarbon, histones H4, H3 and H2B were nitrated at their tyrosine (Tyr) residues [116].

Alternatively, reactive aldehydes generated intracellularly during OS can also modify histones by carbonylation. ROS promotes the formation of α-dicarbonyls or α-oxoaldehydes, including methylglyoxal (MGO) and 3-deoxyglucosone [117,118]. In an interesting study, Ashraf et al. showed that histones that were modified by the action of 3-deoxyglucosone in vitro become less thermostable due to partial unfolding. However, in vivo, this effect would lead to alterations in chromatin structure and gene expression [119,120]. Another study highlighted the role of MGO in histone modification by glycation, which leads to an increase in alpha-helical structures and protein stabilization in vitro [121]. Moreover, Kreuz and Fischle proposed that these differences may be due to different target residues of the different reactive aldehydes or the formation of different end products [89].

ROS can also oxidize lipids to form highly reactive α- and β-unsaturated aldehydes, such as glyoxal, MDA, acrolein, 4-hydroxy-2-nonenal (4-HNE) or 4-oxo-2-nonenal (4-ONE) [122]. It has been described that 4-ONE forms adducts with histones causing inhibition of nucleosome assembly [123]. Furthermore, HNE has been shown to alter histone binding to DNA [124]. Glutathionylation is another modification that can occur in histones due to OS. It refers to the interaction of GSH with the thiol (SH) groups of histones, causing nucleosome instability, gene expression and DNA replication changes [125]. In particular, glutathionylation occurs on cysteine residue 110 in histone variants H3.2/H3.3, producing structural changes in the nucleosome, as shown by circular dichroism studies [126]. OS also alters histone methylation. For example, OS has been shown to inhibit the demethylases of the Jumonji C domain-containing (JmjC) family of proteins, which, like TET, use Fe(II). However, ROS oxidize Fe(II) to Fe(III), causing the attenuation of JmjC histone demethylases [109]. Moreover, it has been proposed that -NO can bind directly to the catalytic iron causing the inhibition of the JmjC family and, therefore, a decrease in demethylation [127,128].

Reduced levels of SAM also cause histone hypomethylation. As explained above, OS inhibits SAM synthesis enzymes; thus, histone methyltransferases (HMTs) are blocked [105,129]. Heavy metals such as nickel, arsenic and chromium impact the posttranslational modification (mainly acetylation, methylation and ubiquitination) of histones [130,131]. The primary mechanism related to this epigenetic modification is the OS generated by these metals, but it has also been observed that nickel can replace iron in its catalytic center in enzymes of the JmjC family. In addition, chromium and/or arsenic can affect ascorbate levels, necessary to reduce Fe and thus attenuate demethylase enzymatic activity in cells. In addition to heavy metals, hypoxic conditions can also catalyze ROS to increase histone methylation marks [130,132].

Several studies have also shown the effect of OS on the balance between histone acetylation and deacetylation. Histone deacetylases (HDACs) can be inhibited by carbonylation, phosphorylation, nitrosylation or glutathionylation because of increased ROS [133] and reactive aldehydes [134,135,136]. Cigarette smoke has been shown to reduce HDAC2 activity by phosphorylation [137,138]. Inactivation of HDAC2 results in its ubiquitination and proteasomal degradation and, consequently, an increase in histone acetylation. In patients with COPD exacerbated by cigarette smoke, an increase in the acetylation of histones H3 and H4 at the *NF-κB* promoter is directly responsible for the deregulation of proinflammatory genes [139]. Similar effects of cigarette smoke have been observed in macrophages, in which sirtuin 1 (SIRT1), an enzyme belonging to HDAC class III, was inhibited by the action of ROS, RNS and reactive aldehydes generated by OS [140]. Subsequently, ROS derived from NADPH oxidase 2 (NOX2) causes a marked elevation in histone H3 acetylation through the activation of histone acetyltransferases (HATs) located within the promoter regions of the Slug gene (*SNAI2*) [141]. However, ROS can also stimulate HDAC expression [142,143] or enhance HDAC activity to decrease acetylation. For example, it has been observed that H_2_O_2_ leads to an increase in lactate dehydrogenase, an enzyme that catalyzes the reduction of lactate to pyruvate, producing NAD+, an essential HDACs cofactor, therefore stimulating deacetylation [144].

OS can also phosphorylate histones. ROS cause double-strand breaks, leading to histone phosphorylation to trigger DNA repair [145,146]. Furthermore, H_2_O_2_ can increase histone phosphorylation in an ATR-dependent manner, independent of the presence of double-strand breaks, suggesting that it fulfills a different signaling function in OS cells [147]. However, the role of OS in histone phosphorylation is controversial since it was found that the catalytic metal ion within protein phosphatases can be oxidized under OS conditions [148]. Because histones are the most abundant chromatin proteins, any change in their quantity, structure or posttranslational modifications will severely impact the overall chromatin structure, influencing gene expression, genome stability and cellular stability replication.

### 3.3. Noncoding RNAs and Oxidative Stress

ncRNAs are epigenetic regulators that can modify gene expression without altering the DNA sequence. ncRNAs are known to be sensitive to ROS and function in accordance with the cellular redox state [149]. Today, transcriptional ncRNAs are classified into small ncRNAs and long ncRNAs (lncRNAs). Small ncRNAs can be classified into microRNAs (miRNAs), PIWI (P-element induced wimpy)-interfering RNAs (piRNAs) and small interfering RNAs (siRNAs) [149]. Various studies have revealed that miRNA-targeted transcription factors (e.g., *c-MYC*, *p53*, *NF-KB*) are redox-sensitive. Therefore, abnormal miRNA expression can be attributed, at least partially, to the deregulation of transcription factors induced by an increase in ROS [150,151]. In a similar study, the transformation of human embryonic lung fibroblast (HELF) cells by chronic exposure to arsenite is mediated by the increased expression of *miR-21* and the activation of the ERK/NF-κB pathway in a ROS-dependent manner [152].

The Drosha-DGCR8 complex regulates the processing of miRNAs from their primary form. ROS have been reported to impact the processing capacity of DGRC8, which is dependent on Fe(III) for its action and thus enables the generation of pre-miRNAs [151,153]. The pre-miRNAs then translocate from the nucleus to the cytoplasm, undergoing Dicer processing. ROS inhibit Dicer activity, hence delaying the production of mature miRNAs, which reflects cell behavior [151]. Oxidative modifications can also influence the shape and size of pre-miRNA in bulges and loops [154]. ROS can also modulate the methylation status of specific promoter regions of miRNA genes, imposing epigenetic regulation on miRNA expression patterns. Hypomethylation of the promoter region of *miR-199a* and *miR-125b* in the presence of ROS is stimulated by the upregulation of *DNMT1* [155]. On the other hand, it was observed that most ROS-responsive miRNAs influence the NRF2 system. For example, cisplatin/OS-induced loss of *miR-34a* leads to the overexpression of *SIRT1*, which is required to activate the NRF2 system [156]. NRF2 and its negative regulator kelch-like ECH-associated protein (KEAP1) constitute a pivotal signaling axis for the regulation of genes involved in redox homeostasis, conferring cellular protection against ROS and tumorigenesis [157]. Under physiological status, KEAP1 acts as an adaptor between NRF2 and the ubiquitination ligase Cullin-3 (CUL3), but upon modification of specific thiols, KEAP1 allows NRF2 to translocate to the nucleus in order to activate the expression of antioxidative, metabolizing and detoxifying genes by binding to the antioxidant response element (ARE) in their regulatory regions. Indeed, class I HDAC enzymes and p65 inhibit ARE-dependent gene expression, in the case of p65 via selective deprivation of the CREB-binding protein (CBP) from NRF2 and recruitment of H3 [158]. Among the genes regulated by the NRF2 system, heme oxygenase (*HO-1*), *UGT1A1*, glutathione S-transferase Mu1 (*GSTM1*) and *NQ01* have been reported to be significantly reduced among human cancer subtypes [159,160]. In consequence, epigenetic modifications in the NRF2 system, including abnormal *KEAP1* promoter methylation, could promote carcinogenesis in organs such as the breasts, lungs, thyroid, prostate, skin, brain, colon and prostate [161,162].

Numerous studies show that air pollutants alter the miRNA expression profile and OS could play an important role in this scenario [163]. For example, exposure to metal-rich particles significantly altered the miRNA expression profile among steel production workers [164]. Furthermore, miRNAs that actively participate in inflammation, endothelial dysfunction and coagulation are significantly altered in subjects exposed to environmental black carbon, organic carbon, PM2.5 and sulfates [165]. Subchronic exposure to cigarette smoke also affected miRNA expression in rat lungs [166]. Authors identified 484 miRNAs, of which 126 were significantly downregulated and 7 were upregulated. Additionally, the lncRNA *HOTAIR* has been shown to be involved in the epithelial-to-mesenchymal transition of human bronchial epithelial (HBE) cells associated with inflammation and OS induced by cigarette smoke extracts [167]. In summary, OS is one of the mechanisms involved in epigenetic alterations induced by environmental factors. Among others, OS generated by heavy metals, PM and endocrine disruptors modifies DNA methylation, causing DNA damage, oxidizing guanine, decreasing SAM concentration and interfering with Fe homeostasis. The production of ROS, RNS and reactive aldehydes also leads to alterations in histone modifications such as acetylation, methylation, phosphorylation, nitration, ribosylation, ubiquitination, sumoylation or glycosylation. In summary, an ROS increase also influences alterations in the expression of ncRNAs by modulating their transcription factors and the enzymes necessary for their production and maturation.

## 4. Microenvironment and Nutritional Influence on the Preservation of Epigenetic Marks

When cells are cultured in vitro, they live and grow in conditions very different from the physiologic environment. Therefore, facing the need to adapt, cells use metabolic routes that allow them to survive and even thrive in the new media. Such adaptations result from epigenetic events [168]. Several studies have demonstrated that cells that proliferate in culture tend to show changes in their epigenetic landscape [169,170,171,172,173]. These changes are not random but, in fact, positively selected for the cell to adapt to the new media and thrive in it [171].

### 4.1. Epigenetic Response to Culture Conditions

Human pluripotent stem cells (hPSC) display a significant change in the pattern of DNA methylation when grown in prolonged in vitro culture [167,172,173]. Even changing the conditions within the culture medium leads to a new period of adaptation and epigenetic modifications (Figure 2). Most of the epigenetic changes induced by culture conditions occur in early stages and are stable and inheritable, even after differentiation events, as has been proven in human embryonic stem cells (hESC) [174,175].

In recent years, hPSC and hESC have been studied as potential therapies for several diseases. Cell therapy, however, faces several important obstacles [176]. One important aspect is that these epigenetic changes could be permanent [169]. Moreover, several of the acquired epigenetic patterns were shown to resemble those of cancer [171]. Weissbein et al. reported changes in the methylation and expression patterns of hPSC, the entity of which strongly correlated with the number of passages in culture. Particularly, it was found that the level of methylation in the cells increases and, in parallel, gene expression decreases in a chronological manner [171]. Additionally, the hypermethylation and downregulation of *TSPYL5* were reported, a pattern common in several types of cancers [177,178] and correlating with cell proliferation [179].

Furthermore, it has been reported that mouse embryonic fibroblasts (MEFs) show a change in the pattern of global lysine acetylation in a matrix-mechanics-dependent manner. MEFs cultured in a stiff hydrogel matrix show a higher general content of lysine acetylation, which might correspond to an open, decondensed, active chromatin [180]. A recent study by Cox et al. showed that culture conditions affect the histone methylation pattern [170]. To characterize H3K4me3 profiles in glioblastoma, U251 cells differed depending on whether the cells were cultured in 2D or 3D microenvironments and were influenced by oxygen levels [163]. Switching from a 2D to a 3D culture was associated with 11,863 differentially methylated loci when cultured in hypoxic and with 11,303 differentially methylated loci when cultured under normoxic conditions. The switch from normoxic to hypoxic caused methylation changes in 1000 regions, and in 1246 regions in 3D cultures. When analyzing the position of H3K4me3 marks within the gene structure, it was observed that in the 3D culture, there was an enrichment in TSS and promoter regions, while in the 2D culture, the distribution was more uniformly distributed in the gene structure. This study supports the idea that the histone methylation landscape is affected by the culture microenvironment. Clearly, it is necessary to study in more detail how mechanotransduction in 3D culture influences cell nuclear phenotypes since the epigenetic organization strongly depends on tissue architecture [181,182,183].

As said above, DNA methylation changes also depend on cellular aging. Consistently, the levels of H3K4me2 and H3K27me2 marks and DNA methylation increased in porcine oocytes cultivated for several generations compared to fresh oocytes. Interestingly, the authors also found that supplementing the medium with melatonin, a known antiaging hormone, attenuated the methylation marks in cultured oocytes to a level similar to that in fresh oocytes [184].

Unfortunately, the changes in ncRNA regulation have not been directly addressed yet in the context of cell culture adaptation. This leaves an unexplored field with a lot of potential, given that many of the experimental models are developed in cell culture and we still lack much knowledge about how much the adaptation process affects the results. However, some initial efforts have shown that it is possible to reprogram cardiac fibroblasts into cardiomyocytes in culture when using a miRNA combination containing *miR-1*, *miR-133*, *miR-208* and *miR-499*. Interestingly, cells showed significantly higher expression levels of matrix metalloproteinases (MMPs) when cultured in a 3D fibrin-based hydrogel as well as a higher rate of reprogramming, suggesting a strong influence of the matrix on the process mediated by an MMPs-dependent mechanism [185].

### 4.2. The Role of Extracellular Vesicles and Additional Cofactors

Cells typically receive external signals such as growth factors, hormones, cytokines, metabolites and other extracellular factors through their sensors (receptors) on the membrane and subsequently commence a cascade of signaling molecules that transmits the signals to the nucleus or other internal organelles. Cells then translate and use these transmitted signals to control many basic functions such as proliferation, growth and differentiation. Furthermore, these signals can trigger posttranslational modification (PTMs) of various proteins, epigenetic changes and chromatin remodeling as an adaptive strategy to the stimuli and, in some cases, this communication can result in abnormal growth and differentiation leading to tumorigenesis. Extracellular vesicles (EVs) that are released by many cell types in different physiological conditions (cancerous and noncancerous), transport signaling biomolecules in the form of proteins, nucleic acids (DNA/RNA), lipids and metabolites that can directly or indirectly regulate a diverse range of cellular processes through long-term epigenetic reprogramming involving DNA methylation, histone modification and posttranscriptional regulation of RNA [186].

Hence, the content of EVs can reflect the cell status and might be clinically relevant to disease progression, diagnosis or assessment of prognosis. For instance, EVs isolated from pancreatic cancer patients were shown to contain genomic double-stranded DNA (dsDNA) bearing mutated *KRAS* and *p53* genes [187]. Microvesicles (EVs of 100–1000 nm in diameter) released from leukemia cells were shown to increase the levels of DNMT3a, DNMT3b and AICDA (a deaminase involved in DNA demethylation) in hematopoietic recipient cells [188]. Moreover, the recipient cells incubated with leukemia-derived microvesicles exhibited a leukemia-like malignant phenotype with increased global DNA methylation levels and hypermethylation of tumor suppressor genes such as *p53* and *RIZ1*, indicating that microvesicles can initiate the malignant transformation of normal hematopoietic transplants through genomic instability.

The role of EVs regulating histone methylation remains elusive. Nevertheless, Schiera et al. discovered that oligodendroglioma cells, but not normal astrocytes, are released in the culture medium EVs containing the differentiation-specific linker histone variant H1° [189]. To be noted, the deregulation of H1° histone expression can be associated with tumorigenesis [189]. Accordingly, H1° is a linker histone variant with a prominent role during terminal differentiation [190,191]. Moreover, studies indicated that tumor-derived factors could inhibit the expression of H1°, which will cause defective dendritic cell differentiation [191]. EVs are quite stable and can be detected in serum. Since ncRNAs are essential regulators of important biological processes and can be packaged in EVs secreted by tumor cells, it follows that EVs in serum can affect the behavior of cells and tissues distant from the primary tumor [192].

Among ncRNAs, miRNAs are probably the most prevalent RNA species found in exosomes, as assessed by deep RNA sequencing accounting for 42.32% of all raw reads and 76.20% of all mappable reads [186,193]. miRNAs are stable in circulation and resistant to RNase digestion in serum [185]. Moreover, miRNA sequences are conserved across species and their expression variations in circulation are linked to different malignancies and disease stages, thus being potential biomarkers [193,194,195,196,197]. In addition, some miRNAs known as “epi-miRNAs” can directly control epigenetic machinery by directly targeting DNA methyltransferases [198]. For instance, *miR-101* represses *DNMT3a* by directly targeting its 3′-UTR and reducing global DNA methylation, with the consecutive expression of the tumor suppressor *CDH1* via hypomethylation of its promoter, hence suppressing lung tumorigenesis [199]. Moreover, *miR-29b* was shown to regulate TET3 levels, which is necessary for the regulation of 5-hmC during memory formation in adult brains [200].

In addition, miRNAs modulate histone deacetylases (HDACs). For instance, the pro-inflammatory activity of *miR-22* was attributed to the posttranslational suppression of HDAC4, influencing the expression of many immune-related genes such as *IL6* and *CD40* [201]. Additionally, *miR-22* was found to be associated with increased macrophage and neutrophil infiltration in the lungs and to be a critical regulator of both emphysema and T(H)17 responses. In addition, *miR-22* was predicted to target *HDAC4*, REST corepressor1 (*RCOR1*) and G-protein signaling 2 (*RGS2*). Overall, the suppression of HDAC4 by *miR-22* promotes neuronal survival and inhibits neurodegeneration in an in vitro model of Huntington’s disease (HD) [202]. In addition, EVs from metastatic cancer cells contain miRNAs contributing to malignancy. For instance, EVs from melanoma cells were shown to carry high levels of prominin-1, which contributed to metastatic progression [203,204,205]. Additionally, the prominin-1-expressing exosomes (prom1-Exo) derived and purified from melanoma and colon carcinoma cells were shown to carry 20 cancer-related miRNAs as well as various prometastatic proteins, including MAPK4K, GTP-binding proteins, CD44, annexin A2 and ADAM10 [206]. In addition, prominin-1-loaded exosomes from melanoma and colon carcinoma cells increased the invasiveness of bone-marrow-derived stromal cells (MSCs) in coculture, highlighting the role of tumor-derived EVs as vehicles to exchange genetic information between tumor and stromal cells, creating a permissive microenvironment for tumor growth and progression. Similarly, the metastatic gastric cancer cell line AZ-P7a, but not the low metastatic AZ-521 cell line, showed high levels of the *mir-let7* family secreted into the extracellular environment via exosomal transport, which induced a prometastatic phenotype in the target cells [207].

The exosomal transport of miR-126 from chronic myelogenous leukemia cells (CML) into endothelial cells (EC) was shown to modulate their motility and adhesion, underscoring the role of exosomal miRNA shuttling in tumor-endothelial crosstalk in the bone marrow microenvironment [208]. Intriguingly, Grange et al. showed that CD105+ MVs, but not CD105− MVs, from renal cancer cells retained their pro-angiogenic properties [209]. In this study, CD105+ MVs were shown to carry multiple proangiogenic mRNAs coding for FGF, VEGF, ephrin A3, angiopoietin1, MMP2 and MMP9, therefore enhancing tumor vascularization. In addition, various cancer-associated miRNAs were detected in CD105+ MVs, including *miR-200c*, *miR-92*, *miR-141* and prometastatic *miR-29a*, *miR-650* and *miR-151*, hence forming a premetastatic niche when administered to a renal tumor cell line. In another study, fibroblast-secreted exosomes were shown to enhance breast cancer cell (BCC) migration by activating autocrine Wnt-planar cell polarity (PCP) signaling, indicating that stromal cells can act to promote cancer progression and metastasis in a specific manner [210]. In addition, co-injection of fibroblasts and BCCs in orthotopic mouse models of breast cancer enhanced metastasis, which was dependent on PCP signaling and, particularly, on the exosome component CD81 from fibroblasts [210]. On the contrary, exosomes isolated from bone marrow mesenchymal stem cell (BM-MSC) cultures positively impacted the behavior of the human breast cancer cell line BM2 in coculture, suppressing the proliferation, decreasing the stem-cell-like surface markers, inhibiting the invasion through Matrigel transwell assay and decreasing the sensitivity to chemotherapy agents such as docetaxel [211]. These effects were attributed to the exosomal transfer of *miR-23b* and the suppression of its target gene, *MARCKS*, that encodes a protein promoting cell proliferation and motility. Furthermore, the same phenotype was observed when *miR-23* was specifically overexpressed in BM2 cells that induced a dormant phenotype by *MARCKS* suppression.

The lncRNA *ANRIL* is known to recruit the polycomb repressor complex 2 (PRC2) through binding to its SUZ12 subunit, thereby mediating H3K27 trimethylation and forming heterochromatin around loci with the tumor suppressor genes *INK4b* (p15), *ARF* (p14) and *INK4a* (p16) [212]. In addition, *ANRIL* can also recruit PRC1 by directly binding to its CBX7 subunit and directing H3K27me3 to stimulate H2AK199 ubiquitination [213]. Like exosomal miRNAs, the content of lncRNAs in EVs might reflect tumor oncogenic hallmarks such as tumor growth, metastasis and response to treatment. Recent studies show that four lncRNAs (*SNHG16*, *ZFAS1*, *OIP-5 AS1* and *ERVK3-1*) were significantly upregulated in exosomes derived from breast cancer tumor cells (TDEs) [214]. Moreover, among them, the exosomal lncRNA *SNHG16* secreted by breast cancer (BC) cells can specifically increase CD73 expression on Vδ1 Treg cells, which are the predominant regulatory T-cell population in BC through the *SNHG16*/*miR-16-5p*/*SMAD5* regulatory axis. HIF-1α-stabilizing long noncoding RNA (*HISLA*) was demonstrated to be transmitted from tumor-associated macrophages (TAMs) to breast cancer cells via secreted exosomes causing metabolic reprogramming by enhancing aerobic glycolysis and apoptosis resistance of breast cancer cells [215]. In addition, the release and transfer of exosomal metastasis-associated lung adenocarcinoma transcript 1 (*MALAT1*) from epithelial ovarian cancer (EOC) cells to recipient HUVEC cells enhanced the expression of angiogenesis-related genes [216], a vital process that was shown to supply tumor cells with nutrients and oxygen for continuous tumor growth and a prerequisite for metastasis.

While emerging shreds of evidence indicate the critical role of exosomal lncRNAs in multiple processes of cancer development, such as hyperproliferation of cancer cells, metastasis, apoptotic resistance, angiogenesis, drug resistance and immunomodulation, the discovery of significant biological information from exosomal lncRNAs can help us to better comprehend and manage the development and progression of cancer. In this context, one of the most promising research areas is the one addressing the association between EVs, oxidative stress and the pathogenesis of multiple diseases, including cancer. Several studies have indicated that when the organism’s homeostasis is altered (for instance, upon an increase in oxidative stress), EVs cargoes radically change and, consequently, so do their downstream effects [217,218,219]. Nevertheless, the relationship between oxidative stress and oxidative modifications inside EVs is still not clearly understood, which limits the full appreciation of the clinical potential of EVs.

Interestingly, a recent study showed that treatment with the oxidative-stress-inducing and chemotherapy agent, doxorubicin (DOX), resulted in increased EV production (DOX-EVs) [220]. Moreover, DOX-EVs exhibited an aberrant morphology with increased levels of 4-hydroxynonenal (4-HNE) adducted proteins, which is a lipid peroxidation product that appears in cell membranes during oxidative stress and is linked to DOX-induced cardiotoxicity. In addition, DOX-EVs exhibited tissue-specific protein isoforms, for instance tissue-specific haptoglobin (Hp) and glycogen phosphorylase (GP) isoforms from the brain (PYGB), skeletal muscle (PYGM) and liver (PYGL). Importantly, pretreatment with a mitochondrial-selective antioxidant-enhancing drug (MnP) significantly reduced the levels of EVs-associated protein-bound 4-HNE when compared to DOX treatment alone. However, another antioxidant-enhancing drug, DRZ, had a weaker effect on the release of EVs than MnP, which might be explained by the higher affinity to mitochondria. Overall, EVs could be used as a predictive tool for the antioxidant reactivity of individual patients upon chemotherapy, as well as a tool to determine the chemotherapy dose that will avoid normal tissue injury.

### 4.3. Nutrigenomics Impacts Epigenomics

In addition to cofactors, cellular metabolism is a key modifier of multiple epigenetic modifications, including DNA methylation, histone methylation and acetylation. Many metabolic intermediates, including SAM, 2-hydroxyglutarate (2-HG), acetyl-CoA, α-ketoglutarate (α-KG), uridine diphosphate-N-acetylglucosamine (UDP-GlcNAc), succinate, fumarate and lactate, provide substrates for epigenetic modifications of histones as well as various nonhistone proteins that have an impact on tumor development. As described earlier, the transfer of the methyl group from SAM to the substrate results in the production of SAH and methylated amino acid. Both SAM and SAH can bind to methyltransferases. However, SAH was reported to bind to some methyltransferases much more tightly than SAM and is a potent inhibitor of many (SAM-)dependent methyltransferases [221,222], which inhibits the conversion of methionine to SAM. Increasing lines of evidence suggest that metabolic changes and histone methylation are highly correlated in cancer cells. For instance, the enhancer of zeste 2 polycomb repressive complex 2 subunit (EZH2) was demonstrated to promote histone H3 lysine 27 trimethylation (H3K27me3), affecting tumor cell metabolism, including carbohydrate, lipid and amino acid metabolism associated with tumorigenesis and cancer development [223]. In another study, EZH2 was shown to promote glycolysis in pancreatic cancer mediated by H3K27me3 and the suppression of *LINC00261*, which acts as a tumor suppressor in pancreatic cancer [224].

Cellular metabolism is oxidative in nature and produces reactive oxygen species as byproducts [48]. Oxidative stress has been proposed as a potential mediator of nutrition-induced epigenetic changes that could be passed on to offspring [225]. Dietary intake of macronutrients, such as protein, carbohydrates and lipids, causes postprandial oxidative stress in several organs and tissues, mainly in vascular-endothelial tissue, adipose tissue, skeletal muscle, nervous tissues and liver and pancreatic β-cells [226,227,228]. Additionally, their metabolism provides intermediates, such as acetate, S-adenosylmethionine (SAM), α-ketoglutarate, uridine diphosphate, (UDP)-glucose, adenosine triphosphate (ATP), nicotinamide adenine, dinucleotide (NAD+) and fatty acid desaturase (FAD), which are utilized for chromatin modification and thus impact epigenetic changes [229].

Micronutrients like vitamins (and minerals) act as cofactors to regulate cell metabolism. These cofactors contribute to the enzymatic activity of epigenetic modifiers and their dietary deficiency has various health consequences affecting physiological growth, immune response, endocrine response and other processes [230,231]. Overall, nutrition acts as an epigenetic modifier to adapt the organism to either the excess or lack of macro- and micronutrients, which eventually impacts the general pathophysiology leading to obesity, cardiovascular diseases or defective growth, infertility and other unhealthy conditions [232,233].

The dietary supplementation of macro- and micronutrients, as well as of natural substances with antioxidant properties, affects the epigenetic signature of key metabolic genes that can prevent oxidative damage and the associated pathophysiological conditions induced by hypercaloric nutrients [229,234]. For instance, a >3-month diet rich in polyphenols (such as the Mediterranean diet) has long-lasting protective anti-inflammatory and antioxidant effects on the cardiovascular system [235]. Moreover, nutrients and natural products, such as amino acids, vitamins and plant/herb-derived polyphenols, can determine long-term adaptative responses to stress by switching the gene expression through epigenetic changes (Figure 3).

The most important adaptive stress response for reestablishing cell homeostasis (thus preventing cell damage and cell death) is macroautophagy, a pathway for lysosomal degradation of oxidized, damaged and redundant cellular components [236]. Autophagy counteracts the accumulation of ROS by p62-mediated sequestration and degradation of KEAP1, which then releases NRF2 that eventually leads to the transcription of antioxidant target genes such as superoxide dismutase, catalase, hemeoxygenase-1 and NAD(P)H:quinone oxidoreductase [237]. The excess of oxidative stress may cause cell damage, which eventually triggers inflammation, thus leading to organ disease. Autophagy also dampens inflammation by removing NLRP3 inflammasome activators [238].

Many bioactive compounds target oxidative stress and inflammation through their byproducts (conventionally considered an agro-industrial waste), which increases the value in the circular economy of products such as extra-virgin olive oil. Its secondary metabolites, including simple phenols, phenolic acids, secoiridoids, flavonoids and lignans, have shown anti-inflammatory effects (decreased C-reactive protein, interleukin-6 and tumor necrosis factor TNFα) in randomized clinical trials with colon cancer patients who consumed them in their diet [239,240]. Importantly, olive mill wastewater extracts drastically decrease the levels of prostaglandin PGE2, lactate dehydrogenase (LDH), nitric oxide synthase (iNOS), cycloxygenase COX2 and TNFα, as shown by ex vivo approaches of rat colon, liver, heart and prefrontal cortex [241]. Further studies have suggested that their protective mechanism of action involves lipid peroxidation and the restoration of glutathione concentrations [242].

At the cellular level, an excess of nutrients (e.g., glucose or fatty acids) overwhelms the oxidative phosphorylation capacity of mitochondria, which then overproduce anion superoxide as a side product [57]. In response to oxidative stress, macroautophagy (particularly mitophagy) is induced as a protective pro-survival action [243]. However, a chronic excess of anion superoxide may eventually inhibit autophagy [244,245]. It is to be noted that autophagy, and particularly mitophagy, is also induced when the cell is deprived of oxygen or nutrients [243].

In this context, it is interesting to note that autophagy is also regulated at epigenetic levels [246,247] and a variety of nutrients and natural dietary products with antioxidant properties also stimulate autophagy via epigenetic mechanisms [248,249,250]. Particularly, plant- and fruit-derived polyphenols and flavonoids, such as, for example, curcumin, resveratrol, quercetin and catechins, exert anti-inflammatory activity and protect against disease through the epigenetic modulation of autophagy [251,252,253,254]. In the next paragraph, we shall discuss in detail the epigenetic mechanisms through which the most relevant nutrients and dietary phytochemicals act.

## 5. Effects of Dietary Nutrients on the Epigenome

Dietary vitamins can modulate the epigenome and therefore have a direct impact on physiological and pathological processes [255,256]. **Vitamin C** is an essential micronutrient that blocks oncogenic transformation induced by carcinogens [257]. The protective role of vitamin C in cancer progression has historically been attributed to its antioxidant activity and the prevention of DNA damage induced by oxidative stress [258]. Vitamin C is suggested to affect the genome activity via regulating epigenetic processes. For example, it has been identified that Fe and 2OG-dependent dioxygenases that catalyze the hydroxylation of methylated DNA and RNA and histones require ascorbate as a cofactor to initiate demethylation [259,260]. It is a cofactor for TET dioxygenases that catalyze the oxidation of 5mC into 5hmC. Ascorbate is also required for the JmjC-domain-containing histone demethylases, serving as a cofactor for histone demethylation. In this manner, vitamin C appears to be a mediator between the genome and the environment. These findings demonstrate an unknown function of vitamin C in regulating the epigenome, which needs a re-evaluation of the functions of vitamin C in human health and disease [257]. Recent studies have shown that vitamin C, by enhancing TET activity, can directly influence DNA methylation levels that alter chromatin structure and the expression of tumor suppressors and DNA repair enzymes [261].

**Vitamin A** (retinol) and its metabolite all-trans retinoic acid reduce DNA methylation by raising the level of TET proteins (which oxidize DNA methylation) and, in doing so, promote stemness [262]. In embryonic stem cells, Vitamin A can displace HDACs from binding to retinoic acid response elements within the promoter of target genes, thus promoting their expression [263]. In neuroblastoma cells, Vitamin-A-induced transcription of the proto-oncogene *RET* is associated with chromatin remodeling and demethylation of H3K27me3 of the enhancer, as well as increased H3K4me3 at the promoter region [264].

**Vitamin B12** is an essential cofactor for the synthesis of methionine (from homocysteine), which is critical for DNA methylation, yet methionine supplement cannot rescue the epigenetic hypomethylation of CpG associated with vitamin B12 deficiency [265]. Deficiency of Vitamin B12 during pregnancy has epigenetic impacts on the next generation, leading to decreased global DNA methylation and increased expression of certain microRNAs such as *miR-221* and *miR-133* [266].

**Vitamin D** is a critical nutrient essential for human health and its deficiency is nowadays a major health problem. It is introduced with food or as a supplement, yet it is largely synthesized in the body upon exposure to solar ultraviolet B radiation. It has been calculated that the vitamin D axis can regulate up to 3% of the genome [267]. The primary epigenetic effects of vitamin D are linked to histone acetylation [268] and DNA methylation [269]. In addition, vitamin D functioning through its receptor VDR is regulated by ncRNAs and, most importantly, VDR itself influences the expression of oncogenic and tumor suppressor lncRNAs [270]. Under the category of **Vitamin E,** saturated tocopherols and unsaturated tocotrienols with ROS scavenging properties are included. α-Tocopherol was reported to increase the methylation of the *miR-9* promoter, a miRNA involved in the control of glycemia [271].

Dietary phytochemicals have antioxidant and anti-inflammatory properties and their prolonged consumption can leave long-lasting epigenetic marks on DNA [272,273] and changes in ncRNAs levels [274,275]. Particularly curcumin, quercetin, resveratrol and EGCG (EpiGalloCathechinGallate), among others, are well-known anti-inflammatory and antioxidant polyphenols and are formidable epigenetic modulators capable of regulating the activity of DNMTs, HATs and HDACs and also the biogenesis of microRNAs [272,274]. Such epigenetic activities have been demonstrated on the basis of the curative potential of these dietary polyphenols in inflammatory diseases [276], cardiovascular diseases [277], metabolic disorders [278] and cancer [253,279,280,281].

## 6. Conclusions and Future Perspectives

The epigenetic regulation of gene expression is a highly regulated process, not less a dynamic mechanism by which our cells respond to both internal and external factors, with these being mechanistically connected. While the epigenetic profile allows the cell to change gene expression signatures in response to stimuli, such as changing conditions from their environment (conventionally referred to as “the exposome”), most of the epigenetic regulators (i.e., enzymes and proteins involved in transcriptional and posttranscriptional modifications) are vastly dependent on the use of a diversity of cofactors from dietary sources. In consequence, a metabolite variation may cause aberrant chromatin structure and, thus, epigenetic deregulation. Furthermore, nutrition in essence is a permanent component of the cellular and tissue microenvironment, bridging an individual’s exposome and genes. In parallel, epigenetics decodes them as an interplay of major impact on the pharmacokinetic properties of bioactive dietary components on their molecular targets.

In this review, we discussed the role of different regulators of chromatin structure, such as DNA methylation and histone-modifying enzymes in syntony with ncRNAs to act upon oxidative stimuli, as well as how diverse nutrients modulate their function in the context of pathophysiological conditions (with a particular focus on cancer). The era of multi-omics approaches provides high-throughput, open platforms for nutrition-related genomics, transcriptomics, proteomics, metabolomics, glycomics and secretomics research. Despite its infancy and our first steps toward real translational approaches, especially in vulnerable populations (for instance, those inevitably exposed to ambient pollution), nutriepigenomics is a growing, promising field that could provide a better understanding and future management of diseases, the risk and outcomes of which are rather unpredictable even among people with similar lifestyles.

## Figures and Tables

**Figure 1 antioxidants-12-00771-f001:**
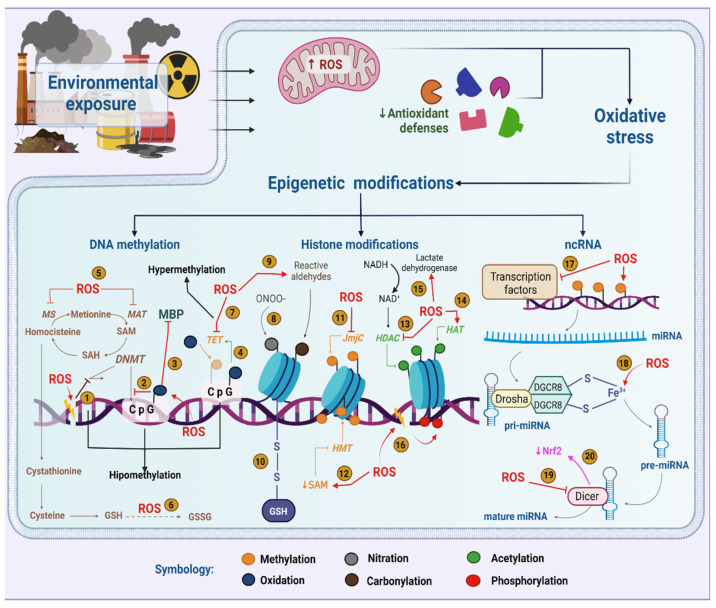
Role of oxidative stress in epigenetic modifications induced by environmental pollutants. ROS potentially interfere with the activity of DNMTs (1), their ability to methylate cytokines (2) or their union with the MBP complex (3). Guanine oxidation makes the methylated cytokine more susceptible to oxidation by TET enzymes (4). In addition, ROS deplete SAM by oxidizing MAT and MS (5) or by using homocysteine to regenerate GSH (6) causing DNA hypomethylation. However, ROS can also inhibit TETs (7). ONOO- nitrates histones (8), while reactive aldehydes modify histones by carbonylation (9). Glutathionylation is a relevant modification in histones due to oxidative stress (10). In addition, ROS either inhibit JmjC demethylases (11) or attenuate HMTs activity by decreased SAM (12). Additionally, ROS inhibit HDACs (13) and stimulate HATs (14). However, ROS could activate HDACs through an increase in NAD+ (15). Histones can be phosphorylated upon DNA damage (16). ROS causes deregulation of transcription factors (17), pre-miRNA synthesis by interaction with Fe^3+^ (18) and Dicer activity inhibition, which impairs miRNA maturation (19) and NRF2 levels (20). Created with biorender.com.

**Figure 2 antioxidants-12-00771-f002:**
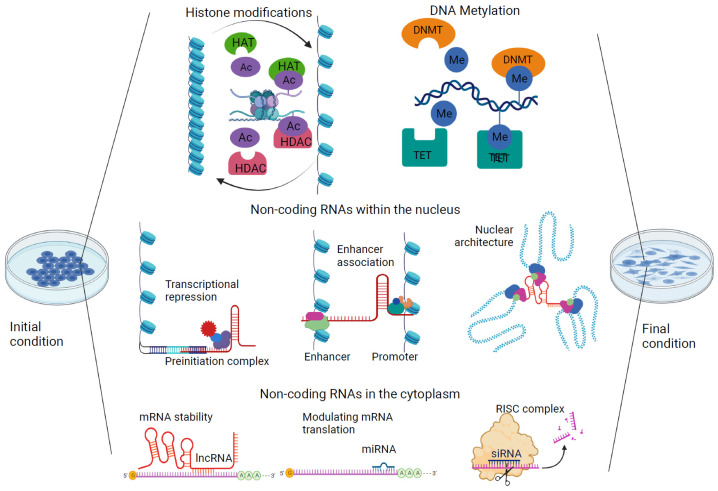
Cells in 2D culture undergo several epigenetic changes that allow them to adapt to their new environment. When cells or tissues are cultured (initial condition), they are affected by the change in environment, which, in turn, stimulates epigenetic modifications that allow the cell to adapt to the new conditions. These changes occur at different levels, histone modification, DNA methylation, the action of lncRNA in the nucleus, which are involved in the regulation of transcription and nuclear architecture, or its action in the nucleoplasm, where it influences RNA metabolism. Created with biorender.com.

**Figure 3 antioxidants-12-00771-f003:**
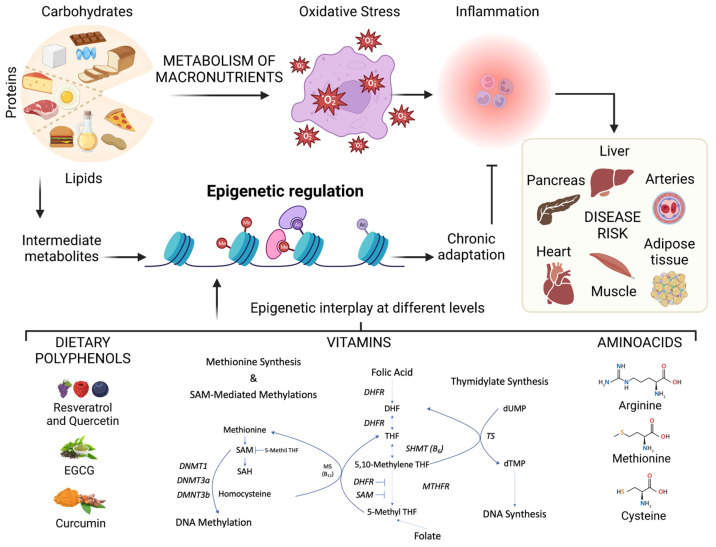
Vitamins linking oxidative stress with epigenetics. The metabolism of dietary macronutrients produces ROS as side-products, which may contribute to inflammation-related diseases. However, intermediate metabolites and other dietary nutrients help the cell to face oxidative stress through epigenetic modulation of genes regulating the stress response. Overview of the metabolism of folic acid for the production of methyl donors (for DNA methylation). Created with biorender.com.

## Data Availability

Not applicable.

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
