# Peer review of "Nutriepigenomics in Environmental-Associated Oxidative Stress"

_antioxidants, 2023, doi:10.3390/antiox12030771_

Round 1

Reviewer 1 Report

In the present manuscript, Rubio and co-workers presented a comprehensive review of the multitude impacts of many different epigenetic and nutritional factors on cellular oxidative stress. The Authors exhaustively explored a plethora of biological pathways and key molecular elements that are implied in oxidative stress, and discussed on how these are modulated and affected by epigenomics and nutrigenomics, thus providing the reader with a clear panoramic view of the topic.

The article is well written, and each reported subtopic is well discussed and referenced. I believe the manuscript deserves publication in Antioxidants only after few minor revisions.

-        Figure 1. The caption of the figure is too long and verbose. I suggest to leave only the first sentence “Role of oxidative stress in epigenetic modifications induced by environmental pollutants” in the caption and move the remaining sentences in the main text, at the beginning of section 3 (as a sort of summary of the section). Moreover, the last sentence of the caption (“This section may be divided…can be drawn”) should be removed.

-        Please replace O2 with O2 throughout the whole manuscript, using subscript for numbers

-        Please replace H2O2 with H2O2 throughout the whole manuscript, subscript for numbers

Author Response

We sincerely appreciate the constructive comments and suggestions from Reviewer  1, which have contributed to the improvement of the current version of our manuscript. In this revised text, we have incorporated all suggestions from Reviewer 1 as detailed below:

  • Figure 1. The caption of the figure is too long and verbose. I suggest to leave only the first sentence “Role of oxidative stress in epigenetic modifications induced by environmental pollutants” in the caption and move the remaining sentences in the main text, at the beginning of section 3 (as a sort of summary of the section). Moreover, the last sentence of the caption (“This section may be divided…can be drawn”) should be removed.

Reviewer 1 is right, figure legend was particularly long in Figure 1, not necessarily contributing to its clarity. In order to preserve the description of 1-20 numbered sequence included in this figure, which is relevant to understand each caption, we summarized the figure legend and removed long terms whose abbreviations are already defined along the manuscript's text. The current version of this figure legend is evidently shorter, and included in Lines 175-185. The same approach was applied to the rest of the figures, for constency purposes. 

  • Please replace O2 with O2 throughout the whole manuscript, using subscript for numbers

We appreciate this observation from Reviewer 1, and we apologize for our previous inconsistency along the text. We have replaced all terms to the correct format.

  • Please replace H2O2 with H2O2 throughout the whole manuscript, subscript for numbers

We appreciate this observation from Reviewer 1, and we apologize for our previous inconsistency along the text. We have replaced all terms to the correct format.

Reviewer 2 Report

In this manuscript the authors focus on reviewing the updated knowledge regarding the impact of epigenetics and its regulation of cellular responses such as oxidative stress.

In my opinion the contribution of nutrigenomics and the role of nutrients should be increased (too many works on epigenetics).

I propose a brief revision with modification of the bibliography.

May I suggest:

-https://doi.org/10.3390/molecules26041072

-https://doi.org/10.3390/toxics10050223

The first indicates that natural compounds also in the circular economy are valid against diseases due to oxidative stress (nutrigenomics). the other because the ingestion of fish containing metals can cause pathologies (epigenomics)

Author Response

We sincerely appreciate the constructive comments and suggestions from Reviewer  2, which have contributed to the improvement of the current version of our manuscript. In this revised text, we have incorporated all suggestions from Reviewer 2 as detailed below:

In my opinion the contribution of nutrigenomics and the role of nutrients should be increased (too many works on epigenetics). I propose a brief revision with modification of the bibliography. May I suggest:

-https://doi.org/10.3390/molecules26041072

-https://doi.org/10.3390/toxics10050223

The first indicates that natural compounds also in the circular economy are valid against diseases due to oxidative stress (nutrigenomics). the other because the ingestion of fish containing metals can cause pathologies (epigenomics)

Reviewer 2 has pointed out two important conceptual aspects of our manuscript which needed further details. The first one, regarding the contribution of by-products from food with antioxidant properties (targeting as well inflammation) to human health. In this regard, we have added a paragraph in Lines 711-722, with four additional references, covering this aspect. We have included here anti-inflammatory and antioxidant targets that are concondantly associated to epigenetic regulation along the manuscript. The second one, regarding the detrimental effect on human health by metals contained in ingested food. To address this point we have included relevant epidemiologic data in Lines 236-241 concerning the oberved increase in cancer risk directly dependent on the concentration of the consumed toxins. Both references suggested by Reviewer 2 were included together with three additional references, we appreciate these suggestions as pivotal to further elaborate our points.

Reviewer 3 Report

The manuscript "Nutrigenomics and epigenomics in environmental-associated oxidative stress" comprises a thorough and well-structured review of the epigenetic mechanisms induced by environmental factors that increase oxidative stress and their potential implications for disease progression. This is a timely review of a relevant topic, contributing to helping understand the long-term impact of environmental pollutants. My only concerns are the following: 

1.      Most of the cited studies discuss how environmental factor-induced ROS/RNS can promote epigenetic changes. Nevertheless, it would also be interesting to discuss how specific epigenetic changes (e.g., methylation of genes related to antioxidant defenses) can result in increased intracellular ROS/RNS levels.

2.      The authors dedicate a section to address how major nutrients derived from the diet may broadly promote epigenetic changes, like DNA methylation or histone modifications. However, they fail to address how such changes may affect specific genes. In this sense, I question the use of the term nutrigenomics in the title of the manuscript, as this anticipates a larger focus on the effects of nutrients on gene expression.

Author Response

We sincerely appreciate the constructive comments and suggestions from Reviewer  3, which have contributed to the improvement of the current version of our manuscript. In this revised text, we have incorporated all suggestions from Reviewer 3 as detailed below:

1.      Most of the cited studies discuss how environmental factor-induced ROS/RNS can promote epigenetic changes. Nevertheless, it would also be interesting to discuss how specific epigenetic changes (e.g., methylation of genes related to antioxidant defenses) can result in increased intracellular ROS/RNS levels.

Reviewer 3 has raised a valid concern about the necessity to cover the endogenous epigenetic mechanisms used by human cells for antioxidant protection, and how this equilibrium is lost therefore increasing intracellular ROS levels. To address this point from the mechanistic point of view, and focused in one the master regulator NRF2 which has been commented in other sections of our manuscript, we have added a paragraph in Lines 406-421, including 6 additional references. We have detailed the endogenous mechanism of gene expression regulation by NRF2 and KEAP1, their most biologically relevant targets, and their dysregulation.

2.      The authors dedicate a section to address how major nutrients derived from the diet may broadly promote epigenetic changes, like DNA methylation or histone modifications. However, they fail to address how such changes may affect specific genes. In this sense, I question the use of the term nutrigenomics in the title of the manuscript, as this anticipates a larger focus on the effects of nutrients on gene expression.

Reviewer 3 has a good point here regarding the conceptual coverage and the focus of our manuscript. By discussing with other co-authors, and also considering similar comments by Reviewer 2, we consider the title "Nutriepigenomics in environmental-associated oxidative stress" to more accurately reflect the contents of our submitted Review. We appreciate this critical observation.